bioengineering/biomaterials/biotechnology

silk scaffold, silver nanoparticles, osteomyelitis

**Authors for correspondence:**
Yixin Shen
e-mail: 972679925@qq.com
Baoqi Zuo
e-mail: bqzuo@suda.edu.cn
Feng Zhang
e-mail: fzhang@suda.edu.cn

†These authors contributed equally to this work.

# Gentamicin-loaded silk/nanosilver composite scaffolds for MRSA-induced chronic osteomyelitis

Peng Zhang[1,†], Jianzhong Qin[1,†], Bo Zhang[2,†], Yi Zheng[3], Lingyan Yang[4], Yixin Shen[1], Baoqi Zuo[5] and Feng Zhang[6,7]

[1]Department of Orthopaedics, [2]Department of Radiology, and [3]Department of Clinical Laboratory, The Second Affiliated Hospital of Soochow University, Suzhou 215004, People's Republic of China
[4]CAS Key Laboratory of Nano-Bio Interface, Suzhou Institute of Nano-Tech and Nano-Bionics, Chinese Academy of Sciences, Suzhou 215123, People's Republic of China
[5]College of Textile and Clothing Engineering, National Engineering Laboratory for Modern Silk, [6]Department of Immunology, School of Biology and Basic Medical Sciences, and [7]Key Laboratory of Stem Cells and Biomedical Materials of Jiangsu Province and Chinese Ministry of Science and Technology, Soochow University, Suzhou 215123, People's Republic of China

FZ, 0000-0001-6348-5816

Methicillin-resistant *Staphylococcus aureus* (MRSA) often induces chronic osteomyelitis and then bone defects. Here, gentamicin-loaded silk/nanosilver composite scaffolds were developed to treat MRSA-induced chronic osteomyelitis. $AgNO_3$ was reduced with silk as a reducing agent in formic acid, forming silver nanoparticles *in situ* that were distributed uniformly in the composite scaffolds. Superior antibacterial properties against MRSA were achieved for the composite scaffolds, without the compromise of osteogenesis capacity. Then gentamicin was loaded on the scaffolds for better treatment of osteomyelitis. *In vivo* results showed effective inhibition of the growth of MRSA bacteria, confirming the promising future in the treatment of chronic osteomyelitis.

## 1. Introduction

Chronic osteomyelitis, a serious clinical disease, usually results in substantial disability. The abuse of antibiotics and cross infection in hospital aggravated the occurrence of chronic osteomyelitis. The most common resistant strains are *methicillin-resistant Staphylococcus aureus* (MRSA). Following the emergence of

superbacteria [1,2], the treatment of chronic osteomyelitis is becoming difficult based on the presently available antibiotics.

The removal of necrosis and infection tissue and then the filling of the cavity are routine procedures for the treatment of chronic osteomyelitis [3]. Polymethyl methacrylate (PMMA), hydroxyapatite and calcium phosphate cement have been used as filling and drug-loading biomaterials. However, PMMA, a non-degradable material with inferior biocompatibility, needs to be removed by re-operation [4] while hydroxyapatite and calcium phosphate failed to release the drugs slowly [5]. Novel biomaterials with better biodegradability, biocompatibility and antibacterial properties are strongly required to fill the gaps.

Although silver ions have broad-spectrum antibacterial and low drug resistance, their clinical application is limited due to the cytotoxicity. As a substitute, AgNPs were developed and showed better physical and chemical stability, excellent bactericidal property and low biological toxicity [6–8]. Silk fibroin (SF) is a natural polymer material with excellent mechanical properties, good biocompatibility and biodegradability, and has been widely used as a scaffold and drug carrier for tissue engineering [9]. Tyrosine residues endow SF with the capacity as stabilizers and reduction agents for the biosynthesis of AgNPs. Various strategies have been developed to prepare AgNPs-loaded SF scaffolds for the treatment of skin infections, refractory wounds and sinusitis [10,11]. However, no attempt was reported to use the scaffolds in chronic osteomyelitis.

Here, AgNPs were formed *in situ* with SF as reducing agents in formic acid and transformed into scaffolds after salt-leaching. Both the biocompatibility and anti-MRSA bacteria capacity of the scaffolds were evaluated *in vitro*. Then gentamicin was loaded on the composite scaffold to improve the feasibility of the treatment of osteomyelitis in a rat model.

# 2. Experimental section

The animal experiments were approved by the Ethics Committee of the Second Affiliated Hospital of Soochow University. All procedures of the animal experiments in this study were performed in accordance with Soochow University Guidelines for the Welfare of Animals.

# 3. Materials and apparatus

The *Bombyx mori* silk was purchased from Soho silk Co., Ltd (Nantong, China). The $Na_2CO_3$, $CaCl_2$, formic acid, $AgNO_3$ and NaCl were supplied by Guoyao Group Chemical Reagent Co., Ltd (Shanghai, China). Gentamycin sulfate was purchased from Renfu Pharmaceutical Co., Ltd (Yichang, China). The MC-3T3 cell line was donated from CAM-SU Genomic Resource Center. ATCC43300 bacteria was donated from the Clinical Laboratory Department of the Second Affiliated Hospital of Soochow University. Dulbecco's minimal essential medium (DMEM), 10% fetal bovine serum (FBS), phosphate-buffered saline (PBS), trypsin, paraformaldehyde, Ketamine and Xylazin were provided by Sigma-Aldrich. The CCK-8 kit was purchased from Dojindo Company, Japan. ALP kit was provided by Shanghai Beyotime Company. Runx2 kit was purchased from Abcam Company, USA.

The optical microscope and confocal laser scanning microscopy (CLSM) were provided by Olympus Company. Scanning electron microscope (SEM) was the Hitachi S-4800 series of Hitachi Company, Japan. X-ray diffractometer was XRD-7000 from Shimadzu Corporation. The Fourier transform infrared spectrometer (FTIR) was the Nicolet 5700 series provided by the US Thermo Electron Corporation. Enzyme-linked immunosorbent detector was supplied by BIO-TEK Company, USA.

## 3.1. Preparation of SF-AgNPs scaffold

The *Bombyx mori* silk was put into 0.05 wt% $Na_2CO_3$, boiled for 30 min and then washed with deionized water three times, the degummed pure silk protein fibre was obtained after drying. Afterwards, the degummed fibres in accordance with 20 wt% concentration were stirred and dissolved in 4 wt% $CaCl_2$ and formic acid (FA) solution for 2 h at 30°C. After the SF was completely dissolved, different concentrations of silver nitrate were added into the solution protected from light for 1 h, and the nanosilver filaments solution was prepared. Then, the NaCl particles with 300–500 particle size were added into the solution and poured into 24-well plates. After the volatilization of formic acid, NaCl and $CaCl_2$ were removed from the soaking water, and finally, the SF porous scaffold containing AgNPs was obtained after freeze-drying (figure 1*a*).

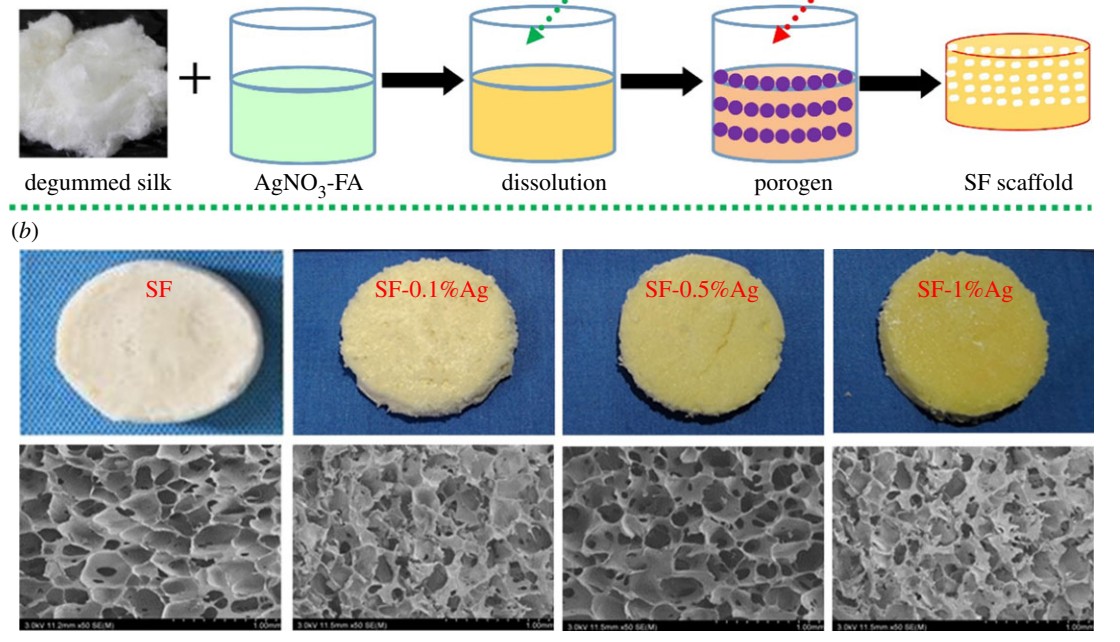

**Figure 1.** Preparation and morphology of SF-AgNPs materials. (*a*) Schematic diagram of the preparation of SF-AgNPs materials. (*b*) The appearance and SEM observation for the SF, SF-0.1%AgNPs, SF-0.5%AgNPs and SF-1%AgNPs.

## 3.2. Characterization of materials

The structural morphology of SF-AgNPs was observed under 10 kV accelerating voltage. Before imaging, the sample was sputter coated with gold under vacuum. The EDX energy-dispersive spectrometer attached to SEM could detect trace chemical elements in materials synchronously.

The crystal structure of the scaffold material was detected by XRD. The diffraction source was copper target (Cu, K$\alpha$ = 1.54056 Å). The scanning region was 2$\theta$ = 5–60°; the voltage 40 kV; the current 300 mA and the scanning rate 0.6° min$^{-1}$.

The interactions and functional groups between SF and AgNPs were evaluated using FTIR. The samples were fully ground and thoroughly mixed with KBr and then assessed in the IR spectra of 400 to 4000 cm$^{-1}$ at room temperature.

## 3.3. Biocompatibility of SF-AgNPs and osteoblasts (MC-3T3)

The human osteoblast-like cell line MC-3T3 cell was selected for compatibility with SF-AgNPs materials. The cell was cultured in DMEM supplemented with 10% FBS. The materials were cut into a circle with 10 mm diameter and 5 mm thickness and sterilized by ultraviolet before the test.

First, we used CLSM to observe the distribution and morphological characteristics of MC-3T3 cells on the materials. The cells were cultured on materials for 5 days, then the cells and material samples were fixed with 4% polyformaldehyde for 15 min. After washing with PBS three times, the cells were stained with DAPI for nuclear fluorescent dye and FITC-phalloidin for F-actin cytoskeleton. After two times of ddH$_2$O washing, the cover glass was placed under the laser confocal microscope to observe and photograph.

According to ISO 10993-5, the cytotoxicity of materials could be detected by the indirect test. Briefly, the materials were soaked in 1 ml of culture medium and incubated at 37°C for 1, 4, 7 and 10 days, and the supernatant was then extracted for detection. Afterwards, 500 μl cell suspension with a density of 5 × 10$^4$ ml$^{-1}$ were seeded onto a 96-well plate for 3 days. Next, the medium was replaced with the extracted supernatant and cultured for 24 h at 37°C. Subsequently, the cell count kit-8 assay and ALP/Runx2 activity were performed.

The CCK-8 assay was used to detect the viability and proliferation of MC-3T3 cell. Twenty microlitres of CCK-8 plus 500 μl DMEM was replaced to each well for 4 h at 37°C. Subsequently, the 300 μl supernatant per well was transferred to new 96-well plates, and the absorbance value was measured using the microplate reader at 450 nm.

ALP/Runx2 activity was tested by the ELISA kit. Briefly, the 0.05% TritonX-100 cell lysate was added for 12 h at 4°C. Finally, according to the manufacturer's protocol of the kit, 20 µl of the suspension was added into the ALP and Runx2 kits in turn. The absorbance was measured at 410 nm.

## 3.4. Antibacterial test

In this study, the MRSA strain selected was ATCC43300. Kirby−Bauer disc diffusion method was used to evaluate the antimicrobial susceptibility of SF-AgNPs materials. Briefly, 100 µl of bacterial suspension was smeared on Mueller Hinton agar plates according to the density of $10^5$–$10^6$ CFU ml$^{-1}$. The samples of different materials (10 mm diameter and 5 mm thickness) were evenly placed in the incubator for 24 h at 37°C. The zone of inhibition was measured three times.

The growth curve method was used to detect the effect of materials on bacterial proliferation. The 50 mg samples of different materials were placed in the flask with 50 ml PBS. Then 5 ml MRSA bacteria solution at a density of $10^5$ ml$^{-1}$ was added, then sealed and cultured on the shaking table at 37°C for 24 h. The 50 µl suspension was collected every 2 h and the colony-forming units were counted after dilution. Bacterial suspension without material was used as a blank control group. The number of colonies at each time point was recorded and the inhibition of bacterial rate curve was drawn.

$$\text{Inhibition of bacterial rate (\%)} = \left[\frac{C_b - C_s}{C_b}\right] \times 100\%,$$

where $C_b$ is the number of culture colonies in the blank control and $C_s$ is the number of cultured colonies in the sample.

## 3.5. *In vivo* osteomyelitis rat model and surgical procedures

According to the results of *in vitro* compatibility and antibacterial activity, we selected 0.5% of AgNPs in SF scaffolds as a scaffold material for *in vivo* studies. Gentamicin combined with scaffold materials was used to treat osteomyelitis in rat models. Pure SF scaffolds and gentamicin combined with SF were used as controls.

Male Wistar rats (260–280 g) were chosen for animal studies. The osteomyelitis model was made as follows: the animals were successfully anaesthetized with Ketamine (60 mg kg$^{-1}$) and Xylazin (3 mg kg$^{-1}$). The proximal tibia was prepared, the limbs were fixed on the operating table, the right proximal tibia was cut through the lateral longitudinal incision, the skin and the surrounding tissue were cut into the tibia and the tibial plateau was exposed. The bone-piercing holes and openings were perforated with the puncture needle in the middle of the plateau. A 2 mm Kirschner needle expanded the hole and opened the bone marrow cavity, then injected into the bone marrow cavity with 5% sodium hyaluronate 0.1 ml and $1 \times 10^8$ CFU ml$^{-1}$ MRSA liquid 0.1 ml. The bone wax was closed to the hole, and the wound was sutured after the saline was washed three times. Antibiotics were not used during and after the operation. One week after the operation, the rats were anesthetized again on the operating table, and the original incision was dissected to re-expose the drilling holes of the tibial plateau. Debridement was performed on the affected limbs and rinsed with saline three times, then the steel brushes and dental burrs were used to eliminate necrotic and infected tissues repeatedly, and rinsed three times with saline again. The blank control group (a) did not have anything placed after debridement ($n = 5$), and the experimental group was implanted with various scaffolds: (b) SF ($n = 6$); (c) SF-AgNPs ($n = 6$); (d) SF + Gentamicin ($n = 6$); (e) SF-AgNPs + Gentamicin ($n = 6$). The wound was cleaned and the incision was closed. Routine feeding was done after the operation. Three weeks after the operation, the rats were sacrificed and bone specimens were taken.

At the time point (the day before osteomyelitis model, the day before surgery, one, two and three weeks after the surgery), the weight of the rats was measured and the white blood cell (WBC) count was detected by the extraction of the caudal vein.

At three weeks after the operation, the X-ray images of the affected limbs were performed. After the integrated tibia was dissected and weighed, the specific infected bone tissues of the tibia upper one-third segment were stained with haematoxylin eosin (HE). Four percent paraformaldehyde was fixed, decalcified and embedded in sections and observed under a microscope.

The remaining part of the tibia was ground and crushed into 10 ml PBS flask, which was flushed with a frequency 1200 r.p.m. for 5 min. The suspension was diluted from $10^2$ to $10^5$, and then colony forming units were counted. Finally, the concentration of bacteria contained in each gram bone was analysed statistically.

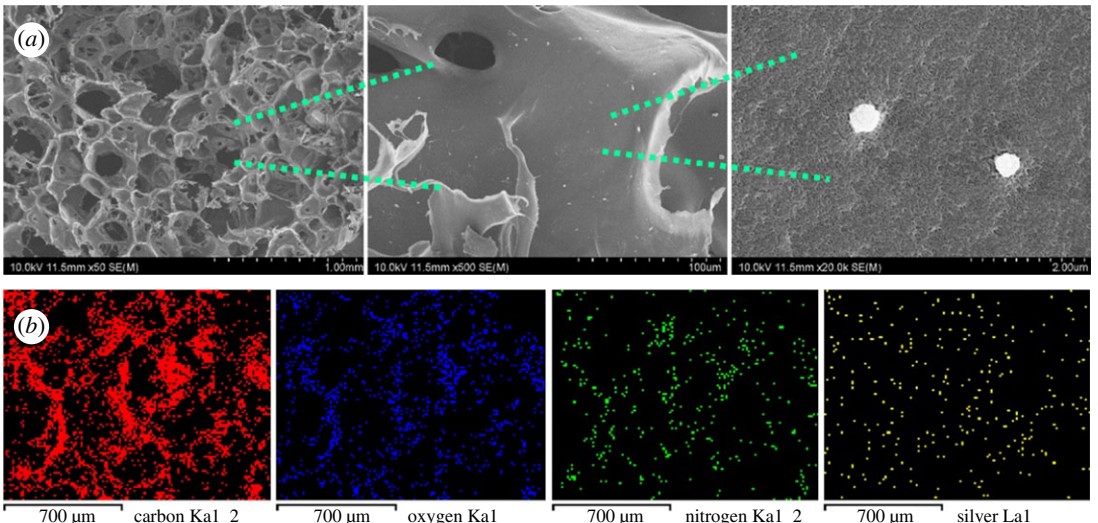

**Figure 2.** SEM observation for the morphology and distribution of AgNPs (*a*) and the EDS spectra of SF-1%AgNPs (*b*).

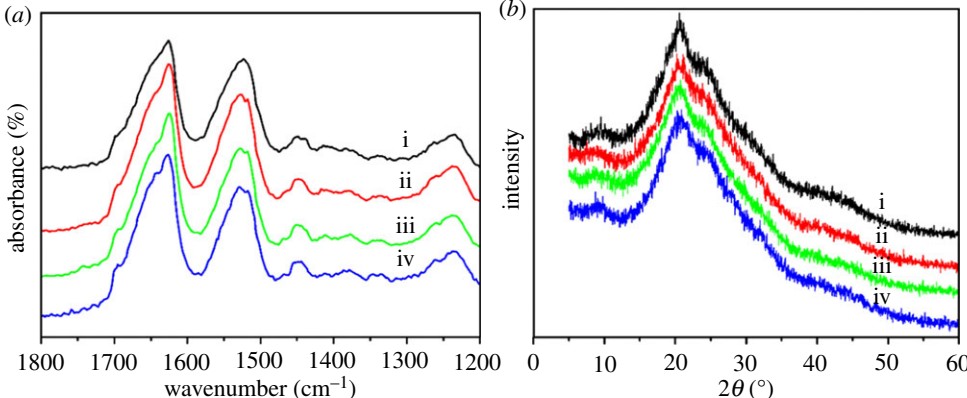

**Figure 3.** FTIR spectra (*a*) and XRD patterns (*b*) of SF (i), SF-0.1%AgNPs (ii), SF-0.5%AgNPs (iii) and SF-1%AgNPs (iv).

## 3.6. Statistical analysis

All the quantitative data were expressed as means ± s.d. All the statistical analyses were performed by the SPSS 16.0. A value of $p < 0.05$ was considered significant. *$p < 0.05$ and **$p < 0.01$.

# 4. Results

## 4.1. Characterization of SF-AgNPs

Figure 1*b* shows the general morphologies of SF three-dimensional scaffolds with different contents of AgNPs. The scaffolds turned yellow gradually with the increasing amount of AgNPs. All the scaffolds showed homogeneous porous structure with the pore size of about 130–460 μm. Although small AgNPs clusters appeared occasionally in the scaffolds with Ag concentration of 0.5% and 1% (figure 2*a*), the EDS spectra indicated the homogeneous distribution of Ag element inside the scaffolds (figure 2*b*). All XRD patterns of the scaffolds showed three typical characteristic peaks at $2\theta = 9.6$, $2\theta = 20.6°$ and $2\theta = 24.1°$, corresponding to the crystal structure of silk II. The results indicated that the introduction of AgNPs had a negligible effect on the transformation of SF in the salt-leaching process, which was similar to the previous report [12]. The FTIR spectra confirmed the stable silk II structure inside all the scaffolds. Two typical peaks at 1618 and 1512 cm$^{-1}$ were observed, which corresponded to silk II structure in amide I and II area (figure 3).

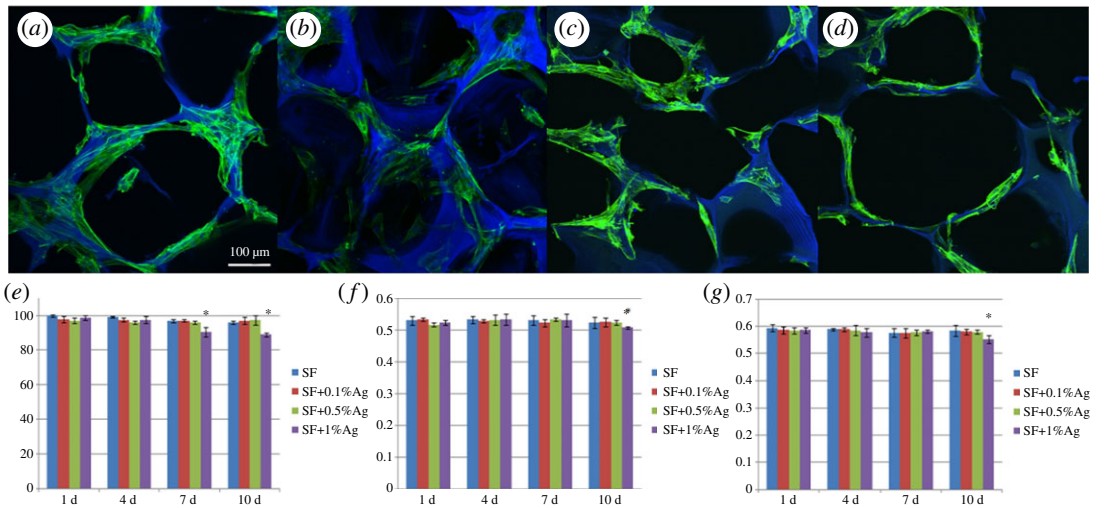

**Figure 4.** Biocompatibility of materials and osteoblasts. The CLSM image of osteoblasts on SF (*a*), SF-0.1%AgNPs (*b*), SF-0.5%AgNPs (*c*) and SF-1%AgNPs (*d*) after 5 days; there was little difference in cell morphology between materials. CCK-8 results (*e*); ALP activity (*f*) and Runx2 activity (*g*). SF scaffolds with 1% concentration AgNPs show time-dependent cytotoxicity (*$p < 0.05$).

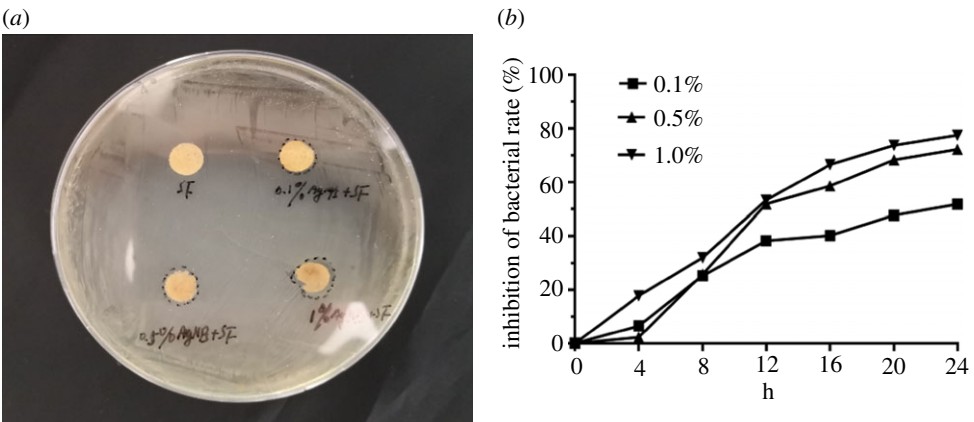

**Figure 5.** Inhibition zones of the SF scaffolds with different concentration AgNPs against MRSA (*a*), the zones of inhibition of SF-0.1%AgNPs, SF-0.5%AgNPs and SF-1%AgNPs were 10.5, 11.4 and 13.2 mm. The inhibition rate converted by bacterial growth curve (*b*) showed the SF scaffolds with 0.5% and 1% concentration AgNPs significantly inhibited bacterial growth.

## 4.2. Cytocompatibility studies

CLSM images (figure 4*a*–*d*) showed the morphology of the osteoblasts cultured on the different scaffolds. After being cultured for 5 days, the cells adhered on the porous scaffolds with elongated morphology, suggesting good cell compatibility of the scaffolds. The viability and function of the cells cultured on the scaffolds were evaluated with CCK-8 assay and ELISA kit (figure 4*e*–*g*). The survival rate of the cells remained above 90% when cultured using scaffolds soaked in culture medium for 10 days, confirming the low cell cytotoxicity of the scaffolds. Although slight inhibition of cell proliferation appeared in the scaffolds with 1% of AgNPs, the cells showed similar growth behaviour on the scaffolds with 0.1% and 0.5% of AgNPs to that on the pure SF scaffolds. The ALP and Runx2 Protein ELISA tests (figure 4*f*,*g*) further confirmed the negligible effect of AgNPs on the cellular functions in the concentration of 0.1 and 0.5%. All the *in vitro* results indicated acceptable biocompatibility of the composite scaffolds for the treatment of osteomyelitis *in vivo*.

## 4.3. Antimicrobial activity

The introduction of the AgNPs endowed the SF scaffolds with good antimicrobial activity. Compared to pure SF scaffolds, the inhibition rate of the scaffolds increased from 51.8% to 72.3% and 77.5% when the Ag concentration was 0.1%, 0.5% and 1%, respectively. Considering suitable antimicrobial activity and better cell compatibility, the scaffolds with 0.5% of AgNPs were used in rat models figure 5.

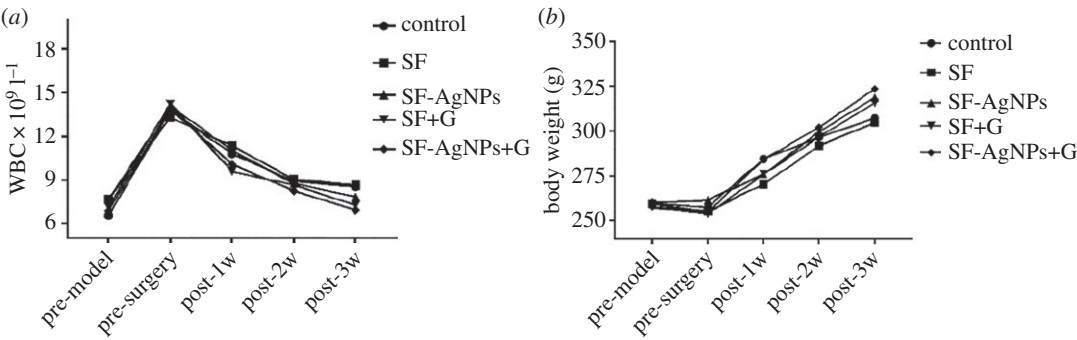

**Figure 6.** The change of body weight (*a*) and white blood cell (*b*) count at the day before osteomyelitis model, the day before surgery, and one, two and three weeks after the surgery.

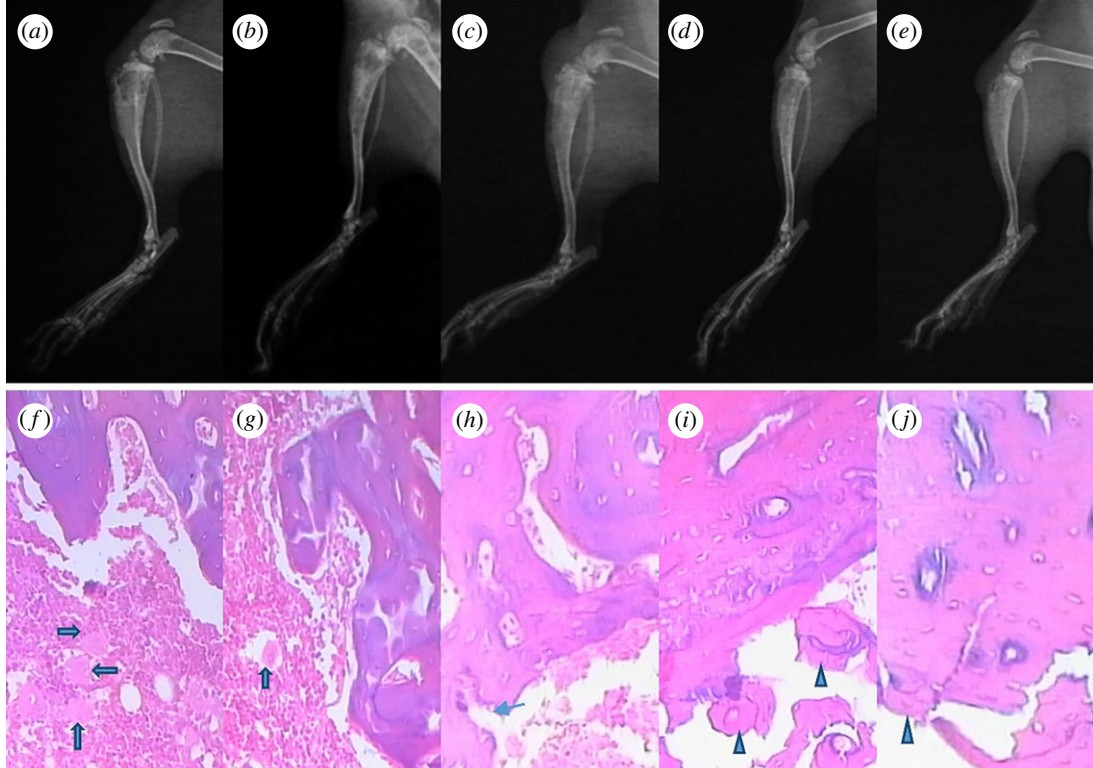

**Figure 7.** Lateral radiographs and histological examination of no implants after debridement (*a,f*), SF (*b,g*), SF-AgNPs (*c,h*), SF + Gentamicin (*d,i*), SF-AgNPs + Gentamicin (*e,j*) after three weeks of treatment. Inflammatory cells (wide arrow), ruptured cortex (narrow arrow) and new bone formation (triangle).

## 4.4. Study of osteomyelitis model

### 4.4.1. General

After the chronic osteomyelitis model formation, the rats without scaffold treatment lost their weight slightly with a significant increase in the WBC. After the debridement and scaffold implantation, the rats restored their normal weights and normal amount of the white blood cells (figure 6*a,b*).

### 4.4.2. Radiology

Significant soft tissue swelling happened for the blank control and the rats implanted with Ag-free SF scaffold. X-ray showed extensive osteolytic cavity, reduced bone mineral density, destruction of cortical bone and dead bone formation for the two samples (figure 7*a,b*). Different to the control,

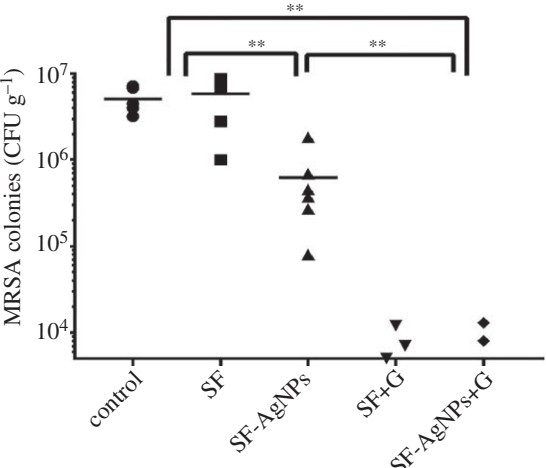

**Figure 8.** Quantitative microbiological analyses of samples between groups. SF-0.5%AgNPs could significantly inhibit the growth of MRSA bacteria in osteomyelitis compared with the blank control group and SF group (**$p < 0.01$).

when the rats were treated with Ag-SF composite scaffolds and gentamicin-loaded composite scaffolds, new bone formed with the increase of bone density (figure 7$c$–$e$).

### 4.4.3. Histology

Histological sections of blank control group and the Ag-free SF implantation group (figure 7$f$,$g$) illustrated significant inflammatory infiltration, abscess formation, osteocyte necrosis, apoptosis and a large number of neutrophils and macrophages, suggesting severe inflammation. When treated with SF-AgNPs-composed scaffolds, the inflammation and abscess significantly alleviated. Only a small amount of inflammatory cell infiltration and periosteal reaction were observed (figure 7$h$). The introduction of gentamicin further restrained the inflammation and induced neovascularization (figure 7$i$,$j$). The inflammatory cells disappeared completely, accompanied by new trabecular bone formation.

### 4.4.4. Microbiology

After three weeks of implantation, the colony count of the affected limb was counted (figure 8). The bacterial amount in the SF-AgNPs group ($6.16 \times 10^5$ CFU g$^{-1}$) was significantly lower than that in the blank control group ($5.14 \times 10^6$ CFU g$^{-1}$) and Ag-free SF scaffold group ($5.85 \times 10^6$ CFU g$^{-1}$) ($p < 0.01$), indicating that the loading of the AgNPs significantly inhibited the growth of MRSA bacteria in osteomyelitis. The bacterial amount was further restrained following the introduction of gentamicin, where the bacterial values for the gentamicin-loaded SF and SF-AgNPs + Gentamicin groups decreased to $4 \times 10^3$ and $3.5 \times 10^3$ CFU g$^{-1}$, respectively. Three of the six cases in SF + Gentamicin group and four of the six cases in SF-AgNPs + Gentamicin group inhibited the bacterial growth completely.

## 5. Discussion

As antimicrobial agents, silver ions exhibited high efficiency, broad-spectrum activity and non-drug resistance. Silver ions had been used to treat chronic osteomyelitis for over 40 years [13], but the toxicity of silver ions to tissues limited their clinical applications. The emergence of AgNPs opened the possibility of silver antibacterial materials in the treatment of osteomyelitis. Different biological scaffolds such as L-polylactic acid, chitosan, calcium alginate and acrylic bone cement, were used to load AgNPs to treat periprosthetic infection and osteomyelitis [14–16].

In our study, a facile one-pot approach was developed to construct porous SF scaffold containing AgNPs, which was superior to previous two-step process where Ag nanoparticles were synthesized and dispersed into the polymer solution [17,18]. The reduction of reducing agent and time as well as better dispersion of AgNPs in the matrices offered the scaffold promising future in different applications [19,20]. Here, FA and SF were used as reducing agents, solvents and matrices simultaneously. SF was also used to stabilize and disperse the formed AgNPs in FA solvent, resulting in homogeneous dispersion of AgNPs in the matrix.

Compared to silver ions, the AgNPs show better physico-chemical stability, super sterilizing and extremely low biotoxicity due to their quantum, small size effect and maximum specific surface area [21]. The antibacterial activity of AgNPs came from direct destruction of biofilm structure, inhibition of bacterial DNA replication, interruption of intracellular signal transduction, inhibition of dehydrogenase activity in the respiratory chain and the production of oxygen free radicals [22–24]. Besides the direct killing effect on bacteria, viruses and fungi, AgNPs could also influence immunomodulatory function [25–28]. Therefore, AgNPs could eliminate multiple pathogenic microorganisms, especially MRSA without the production of new resistant strains. Our *in vivo* and *in vitro* assays confirmed the broad-spectrum bactericidal effect of AgNPs, which was similar to the previous studies [29–32].

Local drug and antimicrobial agent delivery was an effective method of improving the curative effect and reducing the recurrence rate of chronic osteomyelitis. As a suitable candidate for bone tissue engineering, SF protein not only facilitated the growth and differentiation of various cells, such as osteoblasts, bone marrow mesenchymal stem cells and adipose stem cells but also stimulated angiogenesis and bone regeneration. However, enhancing the antibacterial activity of SF material remained a key issue. The loading of AgNPs in biomaterials was a feasible option. Considering the dose-dependence of antibacterial activity for AgNPs, choosing a suitable concentration of the AgNPs was critical for medical applications. Here, the preferable concentration of AgNPs in our study remained 0.5%.

*In vivo* osteomyelitis model confirmed that the loading of AgNPs significantly inhibited the proliferation of MRSA bacteria without the compromise of the advantages of SF materials. Further study was necessary to clarify antibacterial behaviours of AgNPs in osteomyelitis model, especially the effect of AgNPs release behaviour on bacteria *in vitro* and *in vivo*. Although our present results verified the possibility of SF-AgNPs material as filling scaffolds and drug delivery carriers simultaneously to treat chronic osteomyelitis, further improvement was required for better therapeutic effect. Optimizing long-term antibacterial properties of the present system was preferred for osteomyelitis. Additional cross-linking of gentamicin on SF scaffolds could increase the sustained-release behaviour, which would realize in our following study. The accumulation and distribution of silver ions in the main organs of rats and their potential toxicity also needed to be evaluated in the future.

# 6. Conclusion

SF-AgNPs composite scaffolds loaded with gentamycin were designed to treat MRSA-induced chronic osteomyelitis. The resulting SF-AgNPs composite scaffolds showed good three-dimensional pore structure, uniform distribution of AgNPs, favourable biocompatibility with osteoblast and efficient antibacterial properties against MRSA. *In vivo* study indicated that the implanting of gentamycin-loaded composite scaffolds exhibited superior anti-infection effect, suggesting their potential application in repairing osteomyelitis-related bone defects.

Data accessibility. Data available from the Dryad Digital Repository: https://doi.org/10.5061/dryad.5b6r521 [33].
Authors' contributions. Y.S., B.Z. and F.Z. designed the study. P.Z. and J.Q. carried out the cell and animal experiment. P.Z. and B.Z. carried out the data collection and statistical analyses. P.Z., Y.Z. and L.Y. interpreted the results and drafted the manuscript. All authors gave final approval for publication.
Competing interests. The authors declare no competing interests.
Funding. Financial support came from the National Natural Science Foundation of China (81271723), Applied Basic Research Project of Suzhou (SYS201622), a project funded by the Priority Academic Program Development of Jiangsu Higher Education Institutions (PAPD), and Suzhou Planning Project of Science and Technology (SYS201732).

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
