## [Reviewer comments · Royal Society Open Science]

Review History

RSOS-182102.R0 (Original submission)

Review form: Reviewer 1

Is the manuscript scientifically sound in its present form?

Yes

Are the interpretations and conclusions justified by the results?

Yes

Is the language acceptable?

Yes

Is it clear how to access all supporting data?

Yes

Do you have any ethical concerns with this paper?

No

Have you any concerns about statistical analyses in this paper?

Yes

Recommendation?

Accept as is

Comments to the Author(s)

I have read through the manuscript. Overall, this is an interesting paper with an aim to fabricate Silk/nanosilver composite scaffolds loaded with gentamicin for treating MRSA-induced chronic osteomyelitis. In this paper, the dissolving solvent of formic acid and silk as reducing agents for in situ conversion AgNO₃ into silver nanoparticles which were uniformly distributed in the resulting composite scaffolds. In vitro and in vivo studies showed the good biocompatibility with osteoblasts, superior antibacterial properties against MRSA, and effectively treat chronic osteomyelitis. The manuscript can be considered for publication after some minor revisions.

1. The language should be checked carefully to avoid clerical errors, such as the word "degumming" in Page 5, and the disagreement of English tenses throughout the manuscript.
2. Although the authors claimed the superior antibacterial properties against MRSA in vitro, and effectively treatment on chronic osteomyelitis in vivo for Silk/nanosilver composite scaffolds, the inhibition zone test could not support it, as shown in Fig.5.

Review form: Reviewer 2

Is the manuscript scientifically sound in its present form?

Yes

Are the interpretations and conclusions justified by the results?

Yes

Is the language acceptable?

Yes

Is it clear how to access all supporting data?

No

Do you have any ethical concerns with this paper?

No

Have you any concerns about statistical analyses in this paper?

No

Recommendation?

Accept with minor revision (please list in comments)

Comments to the Author(s)

In this paper SF-AgNPs composite scaffolds loaded with gentamycin are designed to treat MRSA-induced chronic osteomyelitis. To this end, the scaffolds mainly require showing biocompatibility

with osteoblast, and antibacterial properties against MRSA.

This study is interesting, and the manuscript can meet the standard of the journal. I would like to recommend it for publication if the authors can address the following points, listed on a chronological basis rather than by importance.

Your data in figure 8, does not show a significant difference between SF+G and SF-AgNPs+G. please provide more explanation how addition of AgNPs can improve the results.

One of the major clinical concerns is how this composite scaffolds perform over a long period of time? how long the effect of gentamicin will last in the composite scaffold. Similarly, how long the effect of AgNPs will last in the composite scaffold? Please discuss more the performance of the scaffold over time.

It is claim in the following reference that the optimum pore size is more than 300 μ m, so since the pore size in this study is between 130 and 460 μ m, do you see any significant differences between pore sizes? (Karageorgiou V. Porosity of 3D biomaterial scaffolds and osteogenesis. Biomaterials. 2005) And does 300-micron pore sizes and higher give better results specially with respect to bone formation and angiogenesis.

What would be the clinical effect of pore size?

What would be the clinical effect of inhomogeneous porous structure?

Can AgNPs be used with other natural hydrogels like gelatin and collagen as well?

What percentage of cells exhibits elongated morphology?

Does the present method have a better dispersion of AgNPs compared to the previous similar methods?

Decision letter (RSOS-182102.R0)

07-Mar-2019

Dear Dr Zhang,

On behalf of the Editors, I am pleased to inform you that your Manuscript RSOS-182102 entitled "Gentamicin-loaded Silk/nanosilver composite scaffolds for MRSA-induced chronic osteomyelitis" has been accepted for publication in Royal Society Open Science subject to minor revision in accordance with the referee suggestions. Please find the referees' comments at the end of this email.

The reviewers and handling editors have recommended publication, but also suggest some minor revisions to your manuscript. Therefore, I invite you to respond to the comments and revise your manuscript.

- Ethics statement

- Data accessibility

It is a condition of publication that all supporting data are made available either as supplementary information or preferably in a suitable permanent repository. The data accessibility section should state where the article's supporting data can be accessed. This section

should also include details, where possible of where to access other relevant research materials such as statistical tools, protocols, software etc can be accessed. If the data has been deposited in an external repository this section should list the database, accession number and link to the DOI for all data from the article that has been made publicly available. Data sets that have been deposited in an external repository and have a DOI should also be appropriately cited in the manuscript and included in the reference list.

If you wish to submit your supporting data or code to Dryad (<http://datadryad.org/>), or modify your current submission to dryad, please use the following link:
<http://datadryad.org/submit?journalID=RSOS&manu=RSOS-182102>

- **Competing interests**

- **Authors' contributions**

- **Acknowledgements**

- **Funding statement**

Because the schedule for publication is very tight, it is a condition of publication that you submit the revised version of your manuscript before 16-Mar-2019. Please note that the revision deadline will expire at 00.00am on this date. If you do not think you will be able to meet this date please let me know immediately.

To revise your manuscript, log into <https://mc.manuscriptcentral.com/rsos> and enter your Author Centre, where you will find your manuscript title listed under "Manuscripts with Decisions". Under "Actions," click on "Create a Revision." You will be unable to make your

revisions on the originally submitted version of the manuscript. Instead, revise your manuscript and upload a new version through your Author Centre.

Once again, thank you for submitting your manuscript to Royal Society Open Science and I look

forward to receiving your revision. If you have any questions at all, please do not hesitate to get in touch.

on behalf of Professor Guy Genin (Associate Editor) and Katrin Rittinger (Subject Editor)
openscience@royalsociety.org

Associate Editor Comments to Author (Professor Guy Genin):

Thank you for submitting your best work to the Royal Society. Both expert reviewers shared my enthusiasm about your paper and suggested some minor revisions.

I note that you are correct and "degumming" is indeed an obscure but real word. For example, according to the Oxford English Dictionary, commercial degumming machines could be purchased all the back in the 1880s. Feel free to leave in that word if you desire.

I look forward to reading your revised manuscript.

Reviewer comments to Author:

Reviewer: 1

Comments to the Author(s)

I have read through the manuscript. Overall, this is an interesting paper with an aim to fabricate Silk/nanosilver composite scaffolds loaded with gentamicin for treating MRSA-induced chronic osteomyelitis. In this paper, the dissolving solvent of formic acid and silk as reducing agents for in situ conversion AgNO₃ into silver nanoparticles which were uniformly distributed in the resulting composite scaffolds. In vitro and in vivo studies showed the good biocompatibility with osteoblasts, superior antibacterial properties against MRSA, and effectively treat chronic osteomyelitis. The manuscript can be considered for publication after some minor revisions.

1. The language should be checked carefully to avoid clerical errors, such as the word "degumming" in Page 5, and the disagreement of English tenses throughout the manuscript.
2. Although the authors claimed the superior antibacterial properties against MRSA in vitro, and effectively treatment on chronic osteomyelitis in vivo for Silk/nanosilver composite scaffolds, the inhibition zone test could not support it, as shown in Fig.5.

Reviewer: 2

Comments to the Author(s)

In this paper SF-AgNPs composite scaffolds loaded with gentamycin are designed to treat MRSA-induced chronic osteomyelitis. To this end, the scaffolds mainly require showing biocompatibility with osteoblast, and antibacterial properties against MRSA.

This study is interesting, and the manuscript can meet the standard of the journal. I would like to recommend it for publication if the authors can address the following points, listed on a chronological basis rather than by importance.

Your data in figure 8, does not show a significant difference between SF+G and SF-AgNPs+G. please provide more explanation how addition of AgNPs can improve the results.

One of the major clinical concerns is how this composite scaffolds perform over a long period of time? how long the effect of gentamicin will last in the composite scaffold. Similarly, how long the effect of AgNPs will last in the composite scaffold? Please discuss more the performance of the scaffold over time.

It is claim in the following reference that the optimum pore size is more than 300 μ m, so since the pore size in this study is between 130 and 460 μ m, do you see any significant differences between pore sizes? (Karageorgiou V. Porosity of 3D biomaterial scaffolds and osteogenesis. Biomaterials. 2005) And does 300-micron pore sizes and higher give better results specially with respect to bone formation and angiogenesis.

What would be the clinical effect of pore size?

What would be the clinical effect of inhomogeneous porous structure?

Can AgNPs be used with other natural hydrogels like gelatin and collagen as well?

What percentage of cells exhibits elongated morphology?

Does the present method have a better dispersion of AgNPs compared to the previous similar methods?

Author's Response to Decision Letter for (RSOS-182102.R0)

See Appendix A.

Decision letter (RSOS-182102.R1)

02-Apr-2019

Dear Dr Zhang,

I am pleased to inform you that your manuscript entitled "Gentamicin-loaded Silk/nanosilver composite scaffolds for MRSA-induced chronic osteomyelitis" is now accepted for publication in Royal Society Open Science.

on behalf of Professor Guy Genin (Associate Editor) and Professor Katrin Rittinger (Subject Editor)
openscience@royalsociety.org

Follow Royal Society Publishing on Twitter: [@RSocPublishing](https://twitter.com/RSocPublishing)

Appendix A

March 30, 2019

Dear Editor,

On behalf of my co-authors, we thank you very much for giving us an opportunity to revise our manuscript, we appreciate editor and reviewers for their positive and constructive comments and suggestions on our manuscript entitled “Gentamicin-loaded Silk/nanosilver composite scaffolds for MRSA-induced chronic osteomyelitis”, which are valuable in improving the quality of our manuscript. We have revised the manuscript point by point according to the comments. The modification has been highlighted in red in the paper, and a detailed point-by-point response to the reviewers’ inputs has been provided.

Looking forward to hearing from you.

Thank you and best regards.

Yours sincerely,

Feng Zhang

Soochow University

Response to reviewer 1

1. The language should be checked carefully to avoid clerical errors, such as the word “degumming” in Page 5, and the disagreement of English tenses throughout the manuscript.

Answer: Thanks for your suggestion. The English tenses in the manuscript had been unified.

2. Although the authors claimed the superior antibacterial properties against MRSA in vitro, and effectively treatment on chronic osteomyelitis in vivo for Silk/nanosilver composite scaffolds, the inhibition zone test could not support it, as shown in Fig.5.

Answer: Fig.5 showed that AgNPs had antibacterial efficiency but its effect on bacteria was not enough. Therefore, its combination with antibiotics is a feasible way to efficient inhibit bacterial growth, as previous report [Bioinspired and Biomimetic AgNPs/gentamicin-embedded Silk Fibroin Coatings for Robust Antibacterial and Osteogenetic Applications.]. In this paper, the in vivo experiment also confirmed that the combination of AgNPs and gentamicin was effective in treating MRSA-induced chronic osteomyelitis.

Response to reviewer 2

1. Your data in figure 8, does not show a significant difference between SF+G and SF-AgNPs+G. please provide more explanation how addition of AgNPs can improve the results.

Answer: The antibiotic, such as gentamicin, was considered as the mostly effective methods for chronic osteomyelitis. As expected, the application of SF+G and SF-AgNPs+G all achieved satisfied results. However, the problem of the solely use of antibiotics was the recurrence of chronic osteomyelitis. For this, the combination of antibiotic and AgNPs would be an alternative choice because the long-term release of Ag⁺ by AgNPs that could provide a antimicrobial microenvironment, thus restraining bacterial regrowth.

2. One of the major clinical concerns is how this composite scaffolds perform over a long period of time? How long the effect of gentamicin will last in the composite scaffold. Similarly, how long the effect of AgNPs will last in the composite scaffold? Please discuss more the performance of the scaffold over time.

Answer: The reviewer raised a very good question. The release behavior of gentamicin and AgNPs was important. In this paper, the gentamicin, simply absorbed into the silk scaffolds before implantation, was released quickly and finished within 1 day, aiming to rapid kill bacteria. In contrary, the AgNPs located in the scaffolds would release for a long time, its complete release was closed related to silk degradation, thus proving a long time antimicrobial microenvironment for bone regeneration. Considered the close relationship between the gentamicin and AgNPs release and scaffold degradation, these topics will be studied in our future work.

3. It is claim in the following reference that the optimum pore size is more than 300 μm , so since the pore size in this study is between 130 and 460 μm , do you see any significant differences between pore sizes? (Karageorgiou V. Porosity of 3D biomaterial scaffolds and osteogenesis. Biomaterials. 2005). And does 300-micron pore sizes and higher give better results specially with respect to bone formation and angiogenesis. What would be the clinical effect of pore size?

Answer: The reviewer's opinion was right. The pore size played a critical role in regulating cell behavior and tissue growth. In this study, most of the pore size was about 300 μm , and some smaller pore formed in the pore wall. We prepared the silk scaffolds using the porogen of NaCl with same partical size, forming similar porous structure, so we didn't study the differences between pore sizes. We believed that it was important to study the effect of silk scaffold pore size on the bone formation and angiogenesis, thus preparing the desired scaffold for clinical application.

4. What would be the clinical effect of inhomogeneous porous structure?

Answer: Thanks for your question. The porous structure, including porosity and pore size, played an important role in bone regeneration in vitro and in vivo. It was still unclear that the clinical effect of porous structure, especially inhomogeneous pore, which needed more studies.

5. Can AgNPs be used with other natural hydrogels like gelatin and collagen as well?

Answer: Yes, AgNPs had been used with gelatin and collagen hydrogels in previous reports. “Physicochemical properties of gelatin/silver nanoparticle antimicrobial composite films”, “Development of gelatin hydrogel pads as antibacterial wound dressings”.

6. What percentage of cells exhibits elongated morphology?

Answer: In our cell experiment, almost all the attached cells exhibited elongated morphology, because the unattached cells would be removed during the exchange of the cell medium.

7. Does the present method have a better dispersion of AgNPs compared to the previous similar methods?

Answer: In our opinion, a similar AgNPs dispersion will be achieved using the method of in situ reducing due to the excellent dispersibility of silver ions.